# Relative Strength Does Not Influence the Sticking Region Among Recreational Trained Participants in Squat

**DOI:** 10.3390/jfmk10030321

**Published:** 2025-08-20

**Authors:** Alexander Olsen, Vidar Andersen, Atle Hole Saeterbakken

**Affiliations:** 1Faculty of Teacher Education and Languages, Department of Natural Sciences, Practical-Aesthetic, Social and Religious Studies, Østfold University College, 1757 Halden, Norway; alexander.olsen@hiof.no; 2Faculty of Education, Arts and Sports, Department of Sport, Food and Natural Sciences, Western Norway University of Applied Sciences, 6851 Sogndal, Norway; vidar.andersen@hvl.no

**Keywords:** kinematics, resistance exercise, maximal strength, biomechanics, repetition maximum

## Abstract

**Objectives**: The barbell back squat is one of the most frequently used exercises to improve lower-body strength and power. The aim of this study was to examine the impact of relative strength on the kinematics in the barbell back squat to a 90-degree angle. **Methods**: Forty-six recreationally trained men completed five familiarization sessions over three weeks to ensure proper lifting technique. The participants were tested in a ten-repetition maximum (10 RM), during which barbell velocity, acceleration, vertical displacement, and the time of the pre-sticking, sticking, and post-sticking regions were measured. The participants were then categorized into two groups: (1) the above-median group or (2) the below-median group, to examine whether kinematics were affected by relative strength (10 RM load/body weight). **Results**: The below-median group had a relative strength of 1.37, whereas the above-median group had a relative strength of 1.76. There was a 5.86% non-statistical difference (*p* = 0.052) in vertical barbell displacement between the groups. There were no significant differences between the groups in barbell velocity or lifting time for the whole movement nor differences between the groups for any of the kinematic variables in the pre-sticking, sticking, or post-sticking regions. When combining the data from the two groups, there was a significant weak negative correlation between relative strength and barbell displacement throughout the whole movement. **Conclusions**: These findings suggest that distinct levels of relative strength may not influence lifting kinematics in 90-degree back squats among recreationally trained participants.

## 1. Introduction

The barbell back squat is one of the most frequently used exercises to improve strength and power in the lower body and it is also one of three exercises in powerlifting [1,2]. A back squat is considered successful if the barbell is lowered to the desired depth (usually 90 degrees, parallel or full squat) and then moved upwards again to the extended position [1]. The full squat (i.e., the top of the hip lower than the top of the knees in the bottom position) is recommended for experienced individuals with proper technique and the required technique in powerlifting. However, for athletes with healthy knees (i.e., free of pain and injuries) a more shallow squat (e.g., 90–110-degree knee-flexion angle) is recommended over a full squat due to the increased injury potential to the menisci and collateral ligaments with increasing knee angle (e.g., 130 degrees in full squat) [3,4].

Further, when performing strength exercises (i.e., bench press, squat) and being close to fatigue, a decreased barbell velocity is observed after the first initial maximal upwards velocity [1,5,6,7]. This region is often referred to as the sticking region and is, in most cases, the region where failure occurs [1,7,8]. It has been suggested that the sticking region occurs as a result of poor mechanical force position, which reduces the capacity to generate force [1,5,6,8]. Previous studies have examined factors affecting kinematics, including repetitions to fatigue [1,9,10], loading variations [11,12], elastic bands [13], overloading (i.e., >1 RM) [7,12], stance width [14], and barbell placement [14]. Of note, the majority of the literature has examined the kinematics in a bench press, but in the last decade, there has been a growing body of literature examining the barbell squat [1,12,14,15,16]. The most recent study [15] concluded that the sticking region results from reduced force output in the pre-sticking and sticking phases. More precisely, suboptimal internal moment arms and lower muscle activity of the knee and hip extensors caused reduced force output to overcome the extensor moments in the pre-sticking and sticking regions among resistance-trained men [15]. Whether greater strength may overcome these factors is yet to be determined.

Several studies have examined the sticking point and kinematics in resistance exercises [17,18,19]; however, information regarding the influence of relative strength on lifting kinematics in squats is scarce. The majority of the previous studies examining lifting kinematics has examined the barbell bench press [5,6], with a handful of studies examining squats [12,14,15,16]. The studies examining the barbell squat have only included one sample size (typically those who are resistance trained or powerlifters) and examined muscle activity, barbell and joint kinematics, repetitions to failure, and barbell position [12,14,15,16]. However, to the authors’ knowledge, the influence of relative strength (i.e., lifted load relative to body weight) on the sticking region has not been examined in squats, though it has been studied in bench presses [20,21]. Saeterbakken et al. [20] showed altered kinematics based on different relative strength levels in the barbell bench press. More specifically, the two groups with the highest relative strength level showed a lower barbell velocity in the sticking region than the lowest relative strength level. The authors of [20] speculated that the differences between strength levels in the sticking region was caused by a different transition of elastic energy between the descending and ascending phases of the barbell. Mausehund and Krosshaug [21] compared the bench press technique and joint kinematics between powerlifters (relative 1 RM strength: 1.55) and recreational lifters (relative 1 RM strength: 1.33). The powerlifters applied a different barbell path, which was displayed by a lower peak elbow net joint movement and shorter joint range of motion than in the recreational lifters. Importantly, both studies examined the bench press and included participants with extensive resistance-training experience [20,21]. It remains unclear as to whether the same differences would be evident in a group with less resistance-training experience or in examining squats. If recreational trained individuals with different relative strengths apply different squat techniques substantially (e.g., hip- or knee-dominant lifting technique), theoretically, different kinematics would display. Potentially, this could expand the previous knowledge of avoiding training to failure in squats before the proper technique has been established. Therefore, it is of great importance and interest to examine whether relative strength influences the kinematics in the back squat in a group with less resistance-training experience than professionals.

To the authors’ best knowledge, no previous studies have examined how strength levels influence the kinematics during the back squat in beginners. Therefore, the aim of this study was to examine the effects of relative strength on the kinematics in the pre-sticking, sticking, and post-sticking regions in recreational trained males in the barbell back squat to a 90-degree angle. Based on the findings from Saeterbakken et al. [20], we hypothesized that the group with greater relative strength would exhibit a lower barbell velocity and longer lifting time in the sticking region than the weaker group.

## 2. Materials and Methods

### 2.1. Design

A cross-sectional study design was used to examine the velocity, acceleration, and load displacement of the barbell in the pre-sticking, sticking, and post-sticking regions in the barbell back squat. Forty-six participants attended a three-week familiarization period where they conducted a total of five squat (90-degree knee angle) sessions (3 sets × 10 RM) to ensure the proper lifting technique and to establish the true 10 RM load. After the five familiarization sessions, the participants were tested using a 10 RM (6th session). During the 10 RM test, the kinematics were measured using a linear encoder attached to the barbell. Based on the participants’ relative strength (i.e., 10 RM load/body weight), the group was divided in the above-median group or below-median group using the median-split technique [22] to examine whether kinematics were affected by the relative strength.

### 2.2. Participants

Based on the barbell velocity between the groups from Saeterbakken et al. [20], and with an α-level of 0.05 and β-level of 0.80, 23 participants in each group were required. Therefore, forty-six healthy and recreational trained [21,23] students were recruited and volunteered as participants (Table 1). All participants were males without pain or injuries that could impair the maximal effort during testing. To be included, the participants had to be free of pain or injuries, had to have not conducted regular strength training of the lower limbs for the past six months (i.e., less than one training session per week), and had to be aged over 18 years. Before experimental testing, their squat technique was subjectively evaluated (e.g., stable lumbar–pelvic–hip complex, absence of knee valgus) to ensure safety. There was no restriction for upper-body-strength-training experience before participation. The participants were encouraged to maintain normal physical activity (sports, endurance training, upper-body resistance training), but not high-intensity training of the lower limbs during the familiarization period. Forty-eight hours before the experimental session, the participants were instructed to avoid all sorts of training of the lower limbs.

Each participant was provided with both an oral and written explanation of the study procedures and any potential risks involved. All participants gave their written consent before being enrolled in the study. The study was approved by the Norwegian Centre of Research data (ref nr: 39024) and conformed to the University ethical guidelines and the latest version of the Declaration of Helsinki.

### 2.3. Procedures

Before all familiarization and testing sessions, participants performed a standardized warm-up with free-weight squats. This involved completing 20 repetitions at 25% of 1 RM, 10 repetitions at 50% of 1 RM, and eight repetitions at 70% of 1 RM, based on their self-reported 1 RM [24]. The repetitions were conducted in self-selected tempo and to a 90-degree knee angle (see further details below). Each set was separated by a 2–3 min pause. No rating of perceived exertion (RPE) scale or repetition in reserve was used to control the intensity of the warm-up [25].

Prior to the experimental test session, the participants conducted five squat familiarization sessions (3 sets of 10 RM) separated by a minimum of 3 days. After each set, the loads increased or decreased based on the number of successful or unsuccessful repetitions. The load was noted for each individual. The test leaders were present during all sets and approved the proper technique before increasing the load. To ensure safety, two experienced spotters were present during all lifts. Furthermore, 10 RM was chosen due to the recommendation by the American College of Sports Medicine (ASCM) position stand [26] and this has previously been examined for the same population [9,27] and among powerlifters (6–8 RM) [21]. To strengthen the ecological validity of the study, we used the traditional repetition maximum (i.e., repetition to fatigue) to determine the 10 RM load instead of velocity-based cut-off values or the RPE scale. Given the included participants technique, safety, and free-weight squat experience, we used a self-selected lifting tempo and RM to ensure fatigue. During the familiarization period, the participants were instructed to refrain from any additional lower-body strength training.

To measure squat depth, a horizontal band corresponding to each participant’s 90-degree knee angle (measured via goniometer) was positioned behind them. The participants were instructed to touch the band with their glutes before pushing upward during the squat movement [28]. For this group, a 90-degree squat depth has been recommended, to improve sport performance, and during preventative rehabilitation programs [29,30,31]. Furthermore, the ASCM position stand [26] recommends a training load corresponding to an 8–12 repetition maximum (RM) for novice and untrained individuals. Accordingly, and since the sticking region occurs using high loads (>85% of 1 RM) or multiple repetitions to fatigue [17,18], we examined a 10 RM in the present study. Furthermore, van den Tillar et al. [32] demonstrated a sticking region of 4–10 com above the lowest barbell position in squats with a minimum depth requirement of lower than 90 degrees in the knee joint. In theory, conducting a 90-degree squat would target the sticking region in the present study.

The participants used their preferred foot width, where the distance between the feet (heels and big toes) was measured and used in familiarization and testing sessions. The participants could either wear footwear or not but could not alter this after the first familiarization session. Knee wraps and belts were not allowed. To minimize the risk of injury, the loads were consciously lowered at a controlled speed in the descending phase. In the ascending phase, the participants were instructed to lift the barbell as fast as possible. The participants were instructed to maintain a natural sway in their lower back. Between the repetitions, only small pauses were allowed. Positive encouragement was given during the final repetitions.

If 10 repetitions were completed in the test session, the loads were increased until the participants and the test leaders agreed that the load was the true 10 RM. The 10 RM loads were determined using 1–3 attempts. The participants were given a 3–5 min rest period between each attempt.

### 2.4. Measurements

A linear encoder (ET-Enc-02, Ergotest Innovation AS, Porsgrunn, Norway) sampling at 200 Hz with a 0.0019 mm resolution was attached to the barbell to analyze kinematics. The encoder was attached to the barbell and situated beneath the weights. Only the concentric phase was included in the analyses (i.e., the turnover from the descending phase to the ascending phase) to identify the pre-sticking, sticking, and post-sticking regions [20]. The encoder has been validated and proven reliable [33,34,35]. The ascending phase of the ninth repetition was used in the analyses, since the final repetition typically differs too much due to the approaches of failure [7,13].

The Musclelab software V10.2 (Ergotest Innovation AS, Porsgrunn, Norway) was used to identify the pre-sticking, sticking, and post-sticking regions [1,13,36]. A visual inspection of the velocity and displacement curve of the barbell was used to identify the 9th repetition among all participants, and the beginning and end of the ascending phase. The pre-sticking region was identified as the region starting at the ascending phase (i.e., barbell velocity = 0 m/s) to the first peak barbell velocity (V_max1_; Figure 1). The sticking region was defined as the region from the end of the pre-sticking region (V_max1_) to the time point when an increased velocity was observed (V_flat_). The post-sticking region was identified as the region from the end of the sticking region (V_flat_) to the peak velocity (V_max2_). The different regions were manually identified using a time sample of the barbell displacement and the barbell velocity before calculating the acceleration (∆ velocity/∆ time) for each of the three regions. The measurements of time, displacement, and velocity demonstrated an inter-and intra-class coefficient (ICC) of 0.559–0.881 (inter rater) and 0.907–0.988 (intra rater). The use of acceleration instead of velocity was used since acceleration has been proposed as being a better discriminative measure than velocity when examining biomechanics during back squats [37].

Previous studies have not demonstrated differences in barbell velocity in the three regions comparing 1, 3, 6, and 10 RM in a squat [9] or the difference in barbell kinematics between the 5th and the 6th repetition at a 6 RM in a squat [38]. Therefore, we included the 9th and not the 10th repetition as the final repetition is too close to fatigue and differs too much from the other repetitions [10,13].

### 2.5. Statistical Analyses

Statistical analyses were performed with SPSS (IBM Corp. Released 2021. IBM SPSS Statistics for Windows, Version 29.0. Armonk, NY, USA: IBM Corp.). The Shapiro–Wilk test was used to check the normality of the data. Eight parameters were found to deviate from a normal distribution (whole movement: time; pre-sticking region: acceleration, displacement in seconds, percentage displacement, and time; sticking region: acceleration, percentage displacement, and time (*p* < 0.02). An independent *t*-test was used to analyze differences between the two groups (i.e., above-median group and below-median group) for the parametric parameters while the Mann–Whitney U test was used for the non-parametric data. Data are presented as the mean ± standard deviation (SD) and effect size. For the parametric data, effect size is reported as Hedges’ g (g), calculated as the mean difference between the groups and the pooled SD. A g-value of ˂0.2 was considered trivial, 0.2–<0.5 considered small, 0.5–<0.8 considered medium, and ≥0.8 considered large [39]. For the non-parametric data, the product movement r (r) is used to report the effect size. R was calculated by the following equation r = z/√n with z being the z-value from the Mann–Whitney U test and the number of participants. An r value of ˂0.1 was considered trivial, 0.1–<0.3 considered small, 0.3–<0.5 considered medium, and ≥0.5 considered large [39].

To calculate the correlation coefficient between the relative strength and the different parameters, Pearson’s r was used for the parametric variables and Spearman’s rho was used for the non-parametric variables. Values of <0.3, 0.3–0.5, 0.5–0.7, and >0.7 were considered very weak, weak, moderate, and strong, respectively [39]. Statistical difference and correlation were accepted at *p* < 0.05.

## 3. Results

The below-median group had a relative strength (10 RM load/body weight) of 1.37, whereas the above-median group had a relative strength of 1.76. Comparing the regions, the participants showed a reduction in acceleration in the sticking region (0.939 ± 0.495 (m/s^2^) vs. 0.285 ± 0.325 (m/s^2^); *p* < 0.001) compared to the pre-sticking and an increase in acceleration in the post-sticking region (0.285 ± 0.325 (m/s^2^) vs. 0.878 ± 0.343 (m/s^2^); *p* > 0.001).

All data for the different groups in the different phases of the ascending movement are presented in Table 2. The above-median group lifted 0.39 kg (28%, *p* < 0.001, g = 2.61) more per bodyweight compared to the below-median group. There was a 5.86% (2.07 cm) non-statistical difference (*p* = 0.05, g = 0.60) in vertical barbell displacement between the groups. There were no differences between the groups in velocity (m/s) (*p* = 0.52) and time (s) (*p* = 0.70) for the whole movement nor differences between the groups for any of the kinematic variables in the pre-sticking, sticking, or post-sticking regions (*p* = 0.12–0.97).

When combining data from both groups, a weak negative correlation between relative strength and barbell displacement was observed for the entire movement (Pearson’s r = −0.38, *p* = 0.01). There was no other significant correlation between relative strength and the other variables in any parts of the moment (*p* = 0.21–0.72, Pearson’s r = −0.19–0.13 and Spearman’s rho = −0.11–0.10).

## 4. Discussion

The aim of this study was to examine the effects of relative strength on the kinematics in the pre-sticking, sticking, and post-sticking regions in the barbell back squat to a 90-degree angle. The main finding of the present study was that relative strength had non-significant small-to moderate effects on the lifting time, barbell velocity, vertical displacement, or acceleration. For the whole ascending movement, no differences in the pre-sticking, sticking, or post-sticking regions in back squat were observed despite a 28% difference in relative strength between the two groups. Combining the groups, no correlations were observed between relative strength and the kinematic variables for any phases of the movement, with exceptions of a weak negative correlation between relative strength and barbell displacement for the whole ascending movement.

All participants showed a reduction in acceleration in the sticking region, but not a clear reduction in barbell velocity, which has been displayed in both the squat [9,12,13,16] and bench press [7,20,40]. The lack of reduced barbell velocity was likely due to the 90-degree knee angle. The typical velocity profile displayed in the sticking region was more of flat region (i.e., no reduction in velocity; see Figure 1). Of note, a similar force profile (i.e., a flat curve) was demonstrated by van den Tillaar et al. [32], comparing dynamic and isometric squats. Importantly, in the dynamic squat conducted in the study by van den Tillaar et al. [32], the minimum depth requirement was a 90-degree knee angle or lower if possible. From the lowest barbell position, van den Tillaar et al. [32] demonstrated a lower force output for the next 15 cm of the vertical barbell displacement. Theoretically, the 90-degree knee angle in the present study should be within the sticking region demonstrated in previous studies [9,12,13,16]. However, during a 90-degree squat, the patellar tendon moment arm is approximately 50% compared to a full squat [41], which may potentially influence the barbell velocity. The lack of a reduction in barbell velocity was most likely to be a result of the 90-degree knee angle. Still, a similar average velocity of the barbell to that in the present study has been reported in studies examining half, parallel, and full squats [42,43]. Of note, both studies [42,43] reported the average velocity of the whole movement and not the pre-sticking and post-sticking regions. Therefore, and given the few comparable studies, the present finding should be interpreted with caution.

It has been suggested that stronger lifters have the ability to exploit the elastic potential from the descending phase, causing a displacement of the force–velocity curve compared to weaker lifters [44]. However, the present study does not support these speculations. It is possible that some of the elastic potential was gradually reduced with the numbers of repetitions conducted or the breakpoint in the force–velocity curve located at very high forces [44]. Several mechanics and adaptions have been suggested to favor stronger participants (e.g., increased neural drive, improved muscle coordination strategies, greater muscle cross-section and/or architecture, single-fiber mechanics or changes in mechanical characteristics of the muscle-tendon complex [45]). Unfortunately, none of these potential mechanics were examined in the present study. However, one could speculate that greater strength would cause a lower barbell velocity in the sticking region due to the increased tolerance of lifting loads with a low velocity [46]. This would potentially display differences in the lifting time in the sticking region between the two groups. Similar lifting kinematics between the two groups in all lifting phases and when analyzing the whole ascending phase could be caused by a low number of participants demonstrating a negative, but only reduced, acceleration in the sticking region. In comparison, Saeterbakken et al. [20] compared bench press kinematics between those with different relative strength levels and showed the longer time spent in each lifting phases with increasing relative strength level. This could be a result of an improved tolerance to continue generating force despite a reduced acceleration of the barbell. Of note, a negative acceleration in the sticking region has been displayed in previous studies examining the kinematics in squats [1,9,12]. However, all of these studies examined a deep squat (i.e., the top of the hip lower than the top of the knees in the bottom position) in comparison to the 90-degree squat examined in the present study.

Even though the cause of the sticking region is not fully known, poor mechanical force generation as a result of the muscle lengths in this region has been suggested [1,5,6]. Therefore, the short moment arms of the muscles involved in performing the 90-degree squat could explain the findings of the present study. For example, the occurrence of the sticking region may be the result of a weak position in the length–force characteristics observed in both squats [15,32] and bench presses [36]. Still, in the study by van den Tillar et al. [1], only 10 out of the 15 resistance-trained participants displayed a sticking region in 6 RM full squats. It is therefore possible that different ascending strategies may explain why some but not all participants had a sticking region in the study by van den Tillaar and colleagues [1] and the present study. For example, different hip angels in the lowest position (i.e., a more forward position vs. a straighter upward position) could cause different level arms, thereby altering the mechanical force generation. In a corresponding manner, an ascending phase driven by a knee extension before a hip extension or vice versa may have resulted in these individual variances in the sticking region observed in the present study. Importantly, these hypotheses were not examined in the present study, but further studies should examine this.

Of great interest is that van den Tillar et al. [15] examined the isometric force output during 10 barbell heights (every 6th cm), simulating the ascending phase of a full squat and compared the force output with 1 RM among resistance-trained men. The lowest isometric force output occurred in the lowest position before the force slightly increased, but not significantly before passing the post-sticking region. Given these findings from van den Tillar et al. [15], a 90-degree knee angel should, in theory, be sufficient to detect the sticking region with altered kinematics.

The present findings were in contrast to those of previous studies examining the effects of relative strength on kinematics [20,21]. Of note, these studies have examined the bench press [20,21]. Mausehund and Krosshaug [21] showed a substantially altered technique, including a shorter joint range of motion with powerlifters compared to recreationally trained lifters in the bench press exercise. Furthermore, Saeterbakken and colleagues [20] showed differences in lifting time, vertical displacement, and barbell velocity between beginners and advanced lifters. In this study, those in the advanced group was significantly lower in stature than those in the beginners’ group and it was speculated that anthropometrics may have influenced the kinematics of the exercise. The above-median group exhibited a ≈6% non-significant difference in vertical barbell displacement (−2.07 cm, g = 0.60) compared to the below-median group. This difference may be attributable to the 2.5% shorter stature (4.4 cm) in the above-median group (Table 1), potentially implying shorter thigh bones and reduced displacement. However, only the stature was included and not the limb segment lengths, which could potentially explain the findings.

In the present study, a 10 RM was examined. Typically, 1–6 RM loads have been used to examine the sticking region and kinematics [5,6,12,47]. Importantly, a 10 RM is recommended as the load to gain muscle strength and hypertrophy [26]. Furthermore, the 10 RM was examined after a three-week familiarization period (five sessions in total) among participants with no regular strength training for the last 6 months. Despite lifting different external loads, the relative intensity was similar for each participant and between the two groups. The identical relative intensity may explain the lack of differences between the groups in terms of lifting kinematics. Importantly, similar barbell, knee and hip kinematics were showed at a 1, 3, 6, and 10 RM in squats among resistance-trained men [9]. Accordingly, the sticking region (displayed as a reduced barbell velocity and longer lifting time) has been shown in high-intensity loads (>80% of 1 RM) [7,40] or lifting repetitions to fatigue [9,10,46]. Different kinematics may have occurred if higher loads (i.e., fewer repetitions before failure) or more experienced lifters were included in the present study.

We observed a weak negative correlation between relative strength and barbell displacement throughout the whole movement, while there were no other significant correlations between relative strength and the other variables in any parts of the movement. The correlation may have been caused by a shorter stature among the strongest participants. Favorable anthropometrics in powerlifting exercises have been debated for decades. For example, short arms will cause a shorter barbell pathway in the bench press than longer arms. It is therefore possible that a shorter stature may have an advantage (i.e., shorter barbell displacement) in terms of the 10 RM strength. However, limb segment lengths were not measured in the present study and therefore no normalization of the segment lengths could be conducted.

The present study has some limitations: (1) Only an encoder measuring the lifting time, barbell displacement, and velocity was used to examine the potential differences between the two groups. Including 3D data or electromyography (EMG) would have given us a possibility of looking at other variables. However, previous studies have demonstrated decreased quadriceps but increased glute activity with increasing barbell height [15,32], and similar findings would have been be expected in the present study. (2) The participants were only divided into two levels of strength. Dividing the group into several strength levels could have given a more in-depth understanding; however, it would also have reduced the power of the analysis considerably. (3) The participants in the present study were recreational trained males with no regular strength-training experience of the lower limbs for the last six months (i.e., less than one session per week). Therefore, these findings cannot be generalized to other populations or females. (4) Recreational trained participants with limited squat experience were recruited. To ensure adequate and secure lifting technique, a 10 RM in the 90-degree squat was used. However, to be able to compare our findings with previous studies, we should have examined a full squat and fewer repetitions to failure. Still, comparable studies have used similar loading and demonstrated a sticking region [9,21]. (5) Only stature and not limb segment lengths were measured. Differences in limb lengths could potentially explain some of the findings and further studies should include these measurements. (6) The inter- and intra-rater variability of the barbell kinematics demonstrated moderate-to-good (inter) and excellent (intra) reliability. Still, automated methods should be included in further studies. Therefore, the present findings should be interpreted with caution. Finally, and despite the sample size calculation stating that sufficient participants were included, we cannot exclude that a type II error has been made due to the low statistical power.

## 5. Conclusions

The main finding of the present study was that the lifting kinematics in the back squat were not discriminated by different levels of relative strength in recreational trained males not conducting lower-limb resistance training on a regular basis. However, a weak negative correlation between relative strength and barbell displacement was observed. Future studies should investigate the potential differences between a full and a 90-degree barbell back squat on the sticking region.

## Figures and Tables

**Figure 1 jfmk-10-00321-f001:**
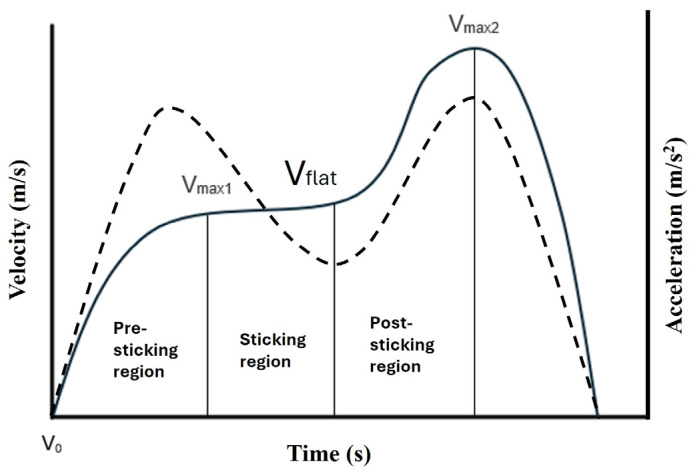
A typical example of the barbell velocity (solid line) and acceleration (dashed line) of the different phases in the ascending phase.

**Table 1 jfmk-10-00321-t001:** The characteristics of the participants.

	All Participants *n* = 46	Below-Median Group *n* = 23	Above-Median Group *n* = 23
Age (years)	22.9 ± 3.4	23.5 ± 4.4	22.3 ± 2.1
Body weight (kg)	78.9 ± 8.5	82.5 ± 9.7	75.6 ± 5.7
Height (cm)	180.4 ± 5.6	182.6 ± 5.4	178.2 ± 5.1
10 RM strength (kg)	120.1 ± 23.0	108.7 ± 12.2	135.9 ± 7.5

**Table 2 jfmk-10-00321-t002:** Kinematic variables in the whole movement and the different regions for the above- and below-median groups. Data presented as mean ± SD.

	Below-Median Group	Above-Median Group	% Difference	*p*-Value	Effect Size	
					**g**	**r**
**Whole movement**						
Relative strength	1.37 ± 0.11	1.76 ± 0.15	28.47	<0.01 *	2.61	
Velocity (m/s)	0.31 ± 0.06	0.30 ± 0.06	−3.22	0.52	0.19	
Displacement (cm)	35.32 ± 3.44	33.25 ± 3.36	−5.86	0.052	0.60	
Time (s)	1.18 ± 0.27	1.16 ± 0.29	−1.69	0.70		0.08
**Pre-sticking region**						
Velocity (m/s)	0.10 ± 0.03	0.09 ± 0.03	−10.00	0.53	0.19	
Acceleration (m/s^2^)	0.90 ± 0.52	0.97 ± 0.49	7.78	0.52		0.14
Displacement (cm)	2.56 ± 1.86	1.81 ± 1.04	−29.29	0.27		0.23
Percentage displacement (%)	7.22 ± 5.21	5.48 ± 3.21	−24.10	0.44		0.16
Time (s)	0.24 ± 0.14	0.19 ± 0.09	−20.83	0.27		0.23
Percentage time (%)	20.36 ± 10.33	16.27 ± 6.76	−20.09	0.12	0.46	
**Sticking region**						
Velocity (m/s)	0.18 ± 0.06	0.17 ± 0.06	−5.55	0.33	0.28	
Acceleration (m/s^2^)	0.30 ± 0.35	0.28 ± 0.32	−6.67	0.94		0.02
Displacement (cm)	2.67 ± 1.54	2.47 ± 1.30	−7.49	0.62		0.10
Percentage displacement (%)	7.51 ± 4.28	7.46 ± 4.14	−0.67	0.97	0.01	
Time (s)	0.15 ± 0.09	0.16 ± 0.09	6.67	0.61		0.11
Percentage time (%)	12.46 ± 5.96	13.36 ± 5.43	7.22	0.60	0.15	
**Post-sticking region**						
Velocity (m/s)	0.52 ± 0.15	0.46 ± 0.10	−11.54	0.15	0.43	
Acceleration (m/s^2^)	0.94 ± 0.31	0.84 ± 0.35	−10.64	0.30	0.31	
Displacement (cm)	24.93 ± 4.83	23.57 ± 3.40	−5.46	0.27	0.32	
Percentage displacement (%)	70.26 ± 9.22	70.95 ± 8.17	0.98	0.79	0.08	
Time (s)	0.52 ± 0.16	0.54 ± 0.18	3.85	0.61	0.15	
Percentage time (%)	44.40 ± 11.68	46.77 ± 9.92	5.34	0.46	0.22	

* = Significant difference (*p* < 0.05), m/s = meter per second, cm = centimeters, s = seconds, m/s^2^ = meter per second squared, g = Hedges’ g, r = correlation coefficient.

## Data Availability

The raw data supporting the conclusions of this article will be made available by the authors, without undue reservation, to any qualified researcher.

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
