# Peer review of "Relative Strength Does Not Influence the Sticking Region Among Recreational Trained Participants in Squat"

_jfmk, 2025, doi:10.3390/jfmk10030321_

Round 1

Reviewer 1 Report

Comments and Suggestions for Authors

Major Concerns

  1. Definition of 'Beginners' and Population Generalizability
  • The term “beginners” is used, but participants were described as physically active and completed 5 training sessions before testing. They may not truly represent untrained individuals.
  • Clarify whether “physically active” includes bodyweight training, sports, or recreational lifting that might affect movement kinematics.
  1. Absence of Full Squat Data Reduces Comparability
  • The study investigates squats to a 90° knee angle rather than full squats, limiting comparison with prior sticking region literature. This is acknowledged but downplayed.
  • Stronger justification is needed for the use of 90° squats (e.g., injury prevention in novices) and how this choice may impact sticking mechanics.
  1. No Direct Assessment of Muscle Activity or Joint Moments
  • While velocity and displacement were measured via a linear encoder, the study draws inferences about joint mechanics and force output without EMG or inverse dynamics.
  • Interpretation of mechanical explanations (e.g., patellar moment arm, internal moment) remains speculative.
  1. Statistical Interpretation and Effect Sizes
  • While group differences were mostly non-significant, the displacement difference (p = 0.05, g = 0.60) is borderline significant and of moderate effect size. This deserves more critical discussion.
  • Likewise, the negative correlation (r = -0.38) between relative strength and displacement is weak but statistically significant. It should not be minimised or overgeneralized.

Specific Comments & Suggestions

Title and Abstract

  • The title is concise and informative.
  • Abstract:

“Strong statistical tendency (p = 0.05)” is awkward phrasing. Say “borderline significant.”

Clarify what “lifting distance” refers to — vertical barbell displacement?

Introduction

  • Provides solid justification and literature grounding.
  • Consider shortening lines 58–69 by condensing the rationale for focusing on beginners.
  • Line 66: “It’s difficult to know...” – revise to academic tone: “It remains unclear...”

Methods

  • Methodology is well-described and ethically sound.
  • Suggestions:

Include exact participant inclusion/exclusion criteria beyond the frequency of training.

Could you explain why the ninth repetition was analysed? Was this consistent for all participants?

Provide reliability or validity data for the encoder device (even if previously published).

State whether limb dominance, footwear, or lifting cues were standardised.

Results

  • Clear and logically structured.
  • Tables are thorough, but consider:

Adding % differences alongside raw data.

Highlighting effect sizes of moderate magnitude even when p > 0.05 (e.g., for displacement).

Discussion

  • Largely balanced, but some issues:

Phrases like “did not influence kinematics” should be softened, given small/moderate effect sizes and correlations.

Could you expand on why the sticking region was observed in some but not all participants?

Acknowledge that kinematics might differ with higher loads or more experienced lifters.

Limitations

  • Well-articulated but could be improved:

Add: sample size may not detect subtle kinematic differences.

Add: no female participants or anthropometric normalisation (e.g., limb segment lengths).

Conclusion

  • Clear and consistent with results.
  • Line 316: Typo — “was that the lifting kinematics…” should be revised for grammar.

Required Revisions:

  1. Clarify the definition and characteristics of "beginners."

  2. Improve interpretation of borderline results (p = 0.05, r = -0.38).

  3. Soften claims around the absence of influence.

  4. Justify the use of 90° squat and its implications more strongly.

  5. Expand discussion of future directions, e.g., exploring joint angles and EMG.

Author Response

We would like to thank the reviewers and editor for your valuable comments and suggestions. We believe they have contributed to improve the quality of the manuscript. Under you will find our point-to-point answers and we have made changes in the manuscript accordingly.

  1. Definition of 'Beginners' and Population Generalizability
  • The term “beginners” is used, but participants were described as physically active and completed 5 training sessions before testing. They may not truly represent untrained individuals.

True, we used a conservative reference (McKay et al., 2022) but we realize “untrained” may not be the best description. Based on a comparable study (Mausehund et al. 2023) and the study from (Santos et al. 2021, DOI: 10.1519/SSC.0000000000000627) defending beginners as able to lift 80 – 120% in squat, we have altered the description of the participants to “recreational trained” throughout the manuscript .

  • Clarify whether “physically active” includes bodyweight training, sports, or recreational lifting that might affect movement kinematics.

Thank you for noticing. We have changed “physically active” to “recreational trained” and expanded the paragraph explaining which activities the participants attended during the study.

  1. Absence of Full Squat Data Reduces Comparability
  • The study investigates squats to a 90° knee angle rather than full squats, limiting comparison with prior sticking region literature. This is acknowledged but downplayed.

We have tried to improve this limitation (please see the discussion) and justified our testing procedures.

  • Stronger justification is needed for the use of 90° squats (e.g., injury prevention in novices) and how this choice may impact sticking mechanics.

We have tried to justify the 90-degree squat (please see methods). Furthermore, we came across a study demonstrating a sticking region the sticking region lasted 4 – 10 cm above the lowest barbell position. The minimum depth requirement was a knee angel of lower than 90 degrees (van den Tillaar et al. 2021). It’s therefore reasonable to assume that a 90-degree squat used in the present study would impact the sticking mechanics. We have expanded the discussion by addressing your comment. Thank you.   

  1. No Direct Assessment of Muscle Activity or Joint Moments
  • While velocity and displacement were measured via a linear encoder, the study draws inferences about joint mechanics and force output without EMG or inverse dynamics.

Yes, we are perfectly aware of these limitations and have highlighted these in the limitation paragraph.

  • Interpretation of mechanical explanations (e.g., patellar moment arm, internal moment) remains speculative.

Yes, as long as we have no data, speculations, hypotheses and discussions of these limitations are all we can include in the manuscript. However, we have expanded the existing discussion of these mechanical explanations. 

  1. Statistical Interpretation and Effect Sizes
  • While group differences were mostly non-significant, the displacement difference (p = 0.05, g = 0.60) is borderline significant and of moderate effect size. This deserves more critical discussion.

Thank you for addressing this. We have expanded the discussion, as suggested.  

  • Likewise, the negative correlation (r = -0.38) between relative strength and displacement is weak but statistically significant. It should not be minimised or overgeneralized.

 We have expanded the discussion, as suggested.

Specific Comments & Suggestions

Title and Abstract

  • The title is concise and informative.

Due to changes in the description of the participants, the title has been changed accordingly.

  • Abstract:

“Strong statistical tendency (p = 0.05)” is awkward phrasing. Say “borderline significant.”

The sentence has been rewritten. Thank you

Clarify what “lifting distance” refers to — vertical barbell displacement?

Thank you for bringing this to our attention. Yes, it is vertical barbell displacement. According to this, changes have been throughout the manuscript.

Introduction

  • Provides solid justification and literature grounding.

We have tried to improve the introduction accordingly and reviewed the literature once again. Two new relevant studies were included.  

  • Consider shortening lines 58–69 by condensing the rationale for focusing on beginners.

We have tried to improve this section. Thank you.

  • Line 66: “It’s difficult to know...” – revise to academic tone: “It remains unclear...

Thank you, the wording has been changed as you suggested.

Methods

  • Methodology is well-described and ethically sound.

Thank you for the comment.

  • Suggestions:

Include exact participant inclusion/exclusion criteria beyond the frequency of training.

Thank you for this comment. We have tried to provide clear inclusion- and exclusion criteria.

Could you explain why the ninth repetition was analysed? Was this consistent for all participants?

Yes, the 9th repetition was used for all participants. The final repletion was excluded as it deviated so much from the “normal” velocity curve which are in line with previous studies (https://doi.org/10.1371/journal.pone.0235555; https://doi.org/10.1080/02640414.2013.803593; https://doi.org/10.1519/JSC.0000000000001178. The authors were present during all testing, and we can guarantee that the true 10RM was tested. We had several failures in the 10th repetitions (?) and to honest; the 10th repetition was in many cases far from beautiful executed in terms of lifting technique.  

References to our choice and clarification of the use of the 9th repetition has been included in the manicurist.

Provide reliability or validity data for the encoder device (even if previously published).

Included, as suggested.

State whether limb dominance, footwear, or lifting cues were standardised.

Limb dominance was not considered, but the choice of footwear was. No standardized cues were used, but positive encouragement and cues were included in the final repetitions. Please see the details included in the methods.

Results

  • Clear and logically structured.

Thank you

  • Tables are thorough, but consider:

Adding % differences alongside raw data.

Included, as suggested

Highlighting effect sizes of moderate magnitude even when p > 0.05 (e.g., for displacement).

Highlighted, as suggested

Discussion

  • Largely balanced, but some issues:

Phrases like “did not influence kinematics” should be softened, given small/moderate effect sizes and correlations.

Agree, and a good point. Thank you. The phrase has been re-written.

Could you expand on why the sticking region was observed in some but not all participants?

From a scientific point of view based on empiric data, no. But I/we can make hypotheses with possible and rational explanations. These explanations have been included in the discussion. Briefly, these explanations have been included:

  • Some participants may have gone deeper than others (only a rubber band provided a stimulus when they were deep enough to start the ascending phase.
  • Different length of anthropometrics especially femur length
  • Different ascending strategies/techniques (i.e., hip driven or knee driven extension) causing different hip angels in the lowest position (forward or more straight up-ward position of the torso).

Acknowledge that kinematics might differ with higher loads or more experienced lifters.

Included in the discussion, as suggested.

Limitations

  • Well-articulated but could be improved:

Add: sample size may not detect subtle kinematic differences.

Included, as suggested.

Add: no female participants or anthropometric normalisation (e.g., limb segment lengths).

Included, as suggested.

Conclusion

  • Clear and consistent with results.

Thank you

  • Line 316: Typo — “was that the lifting kinematics…” should be revised for grammar.

We apologize. The sentence has been revised for grammar and re-written.

Required Revisions:

  1. Clarify the definition and characteristics of "beginners."

Changes as been made, as amended.

  1. Improve interpretation of borderline results (p = 0.05, r = -0.38).

Changes as been made, as amended.

  1. Soften claims around the absence of influence.

Changes as been made, as amended.

  1. Justify the use of 90° squat and its implications more strongly.

Changes as been made, as amended.

  1. Expand discussion of future directions, e.g., exploring joint angles and EMG.

Changes as been made, as amended.

Thank you once again for your insight and valuable comments/suggestions.

Reviewer 2 Report

Comments and Suggestions for Authors

Initially, I would like to thank you the Editors from J. Funct. Morphol. Kinesiol for the opportunity to have reviewed Manuscript ID jfmk-3750006 “Relative strength does not influence the sticking region among beginners in resistance training”. According to information provided by the authors, the main goal of the study was to evaluate the effects of relative strength (independent variable) on the barbell back squat kinematics (pre-sticking, sticking and post-sticking regions) (dependent variables) of beginners. A total of 46 male participants completed the study (cross-sectional). Main findings indicated no influence of the independent variable in the dependent variables. The study seems well conducted and written at a good standard. However, several revisions are required before I can make a final decision on its acceptability for publication. This includes as for example a better/solid justification for the study, sample size calculation, inclusion of intra and inter-rater variability for calculating the dependent measures; avoid talking about trends in results. Below I am sharing specific comments that the authors need to take into account:

L20-21. “There was a strong statistical tendency (p=0.05) towards a 6% shorter lifting distance for the above average group.” – statistical tendency should be noted reported. Please revise the whole results/conclusions in the main text and abstract to make sure that the statements made do not contain speculations

L26-27. “These findings suggest that relative strength does not alter lifting kinematics in back squat to 90-degrees in physical active males.” – I suggest replacing with “These findings suggest that distinct relative strength levels may not influence lifting kinematics in back squat to 90-degrees in beginners in resistance training”

L58-59 “Only one study has examined the influence relative strength on sticking region, but the study examined barbell bench press [19].” - I was curious to know how the authors arrived at this statement that only one study... Did they base their justification on a systematic review or something similar? It is important to state this so that the justification for the study can be much more solid.

L67-69. “Therefore, it is of great importance and interest to examine whether relative strength influences the kinematics in the back squat in a group with little to no training experience.” - Why is it important? Please expand.

L70. The final paragraph of the introduction is not flowing well, and it seems to me that it even presents aspects that should be place in methods

L77-79. “Therefore, the aim of this study was to examine the effects of relative strength on the kinematics in the pre-sticking, sticking and post-sticking regions in beginners when lifting the barbell back squat to a 90-degree angle.” – Please insert hypothesis

L94. “Forty-six healthy and physically active males were recruited and volunteered as participants (Table 1).” – It seems that the number of participants can be sufficient, but it is mandatory to insert sample size calculation in order to confirm this assumption

L97. Please insert if there is maximal age limit to be included in the study; and if this would have affected the results, please provide the corresponding discussion

L98. The authors should have included at least one reference supporting the technique of dividing participants into below and above average. Please insert/comment in the new manuscript version; I had already heard about the median-split technique before but not average

L158-160. “A visual inspection of the velocity curve was used to identify the different regions and if there was any doubt, a second opinion was gathered.” - Why not implement an automatic detection method (it may already exist)? If the authors choose to keep it this way, it is necessary/mandatory to include measures of intra- and inter-rater variability and ensure that this does not compromise the results of the study.

L161-162. FIGURE 1 contains orthographical errors, please revise and replace with a figure with better image resolution

L186-188. “All participants showed a reduction in acceleration in the sticking region, showing a similar pattern throughout the ascending movement.” – This statement has been based on statistical analysis, if not please consider delete

L190. “The above average group lifted 0.39 kg (28%, g = 2.61) more per bodyweight compared to the below average group.” – insert the p-value

L191. “There was also a strong tendency (p = 0.05, g = 0.60) towards a 6% shorter lifting distance for the above average group (2.07 cm).” – do not declare tendencies

L231. In the first paragraph of discussion, I suggest to repeat the main objective before reporting the main findings

L238. “All participants showed a reduction in acceleration in the sticking region.” – it seems that no results with statistical confirmation have been presented to support such statement in the results section. Please revise the discussion as well as the whole manuscript to avoid making statement that are not based on statistical outputs

L317. The first sentence of conclusions section is currently awkward, please revise “The present study was that the lifting kinematics . . .” (sic)

Author Response

We would like to thank the reviewers and editor for your valuable comments and suggestions. We believe they have contributed to improve the quality of the manuscript. Under you will find our point-to-point answers and we have made changes in the manuscript accordingly.

Initially, I would like to thank you the Editors from J. Funct. Morphol. Kinesiol for the opportunity to have reviewed Manuscript ID jfmk-3750006 “Relative strength does not influence the sticking region among beginners in resistance training”. According to information provided by the authors, the main goal of the study was to evaluate the effects of relative strength (independent variable) on the barbell back squat kinematics (pre-sticking, sticking and post-sticking regions) (dependent variables) of beginners. A total of 46 male participants completed the study (cross-sectional). Main findings indicated no influence of the independent variable in the dependent variables. The study seems well conducted and written at a good standard.

However, several revisions are required before I can make a final decision on its acceptability for publication. This includes as for example a better/solid justification for the study, sample size calculation, inclusion of intra and inter-rater variability for calculating the dependent measures; avoid talking about trends in results. Below I am sharing specific comments that the authors need to take into account:

 Thank you for your comments and suggestions. We have tried to improve the justification of the study, including sample size calculation, and changed the discussion about statistical trends and variability of the outcomes.

L20-21. “There was a strong statistical tendency (p=0.05) towards a 6% shorter lifting distance for the above average group.” – statistical tendency should be noted reported. Please revise the whole results/conclusions in the main text and abstract to make sure that the statements made do not contain speculations

Changed, as suggested

L26-27. “These findings suggest that relative strength does not alter lifting kinematics in back squat to 90-degrees in physical active males.” – I suggest replacing with “These findings suggest that distinct relative strength levels may not influence lifting kinematics in back squat to 90-degrees in beginners in resistance training”

Agree, changed as suggested

L58-59 “Only one study has examined the influence relative strength on sticking region, but the study examined barbell bench press [19].” - I was curious to know how the authors arrived at this statement that only one study... Did they base their justification on a systematic review or something similar? It is important to state this so that the justification for the study can be much more solid.

We perfectly understand your point. We have included several search terms and combined them with “AND” and “OR” without identifying several papers. Its wrong to claim the searches have been conducted systematically according to guidelines of a systematic review, but we have made an effort and identified some studies we later decided to leave out DOI: 10.1177/0018720809360801, doi: 10.1515/hukin-2015-0109; doi.org/10.1080/02640414.2023.2172797. Furthermore, I made a call to two leading expert in sticking region and they were not familiar with other studies examining the potential influence of relative strength and sticking point.

Still, we have rewritten the sentence and used the term “to the authors best knowledge” and expanded the Mausehund et al 2023 study which compared powerlifters and recreational trained subjects in bench press.

L67-69. “Therefore, it is of great importance and interest to examine whether relative strength influences the kinematics in the back squat in a group with little to no training experience.” - Why is it important? Please expand.

Thank you for addressing this. We have expanded the paragraph as suggested.

L70. The final paragraph of the introduction is not flowing well, and it seems to me that it even presents aspects that should be place in methods

Yes, we agree and have moved the arguments of procedures to the methods (please see the paragraph “procedures”).

L77-79. “Therefore, the aim of this study was to examine the effects of relative strength on the kinematics in the pre-sticking, sticking and post-sticking regions in beginners when lifting the barbell back squat to a 90-degree angle.” – Please insert hypothesis

Hypothesis has been included, as suggested.

L94. “Forty-six healthy and physically active males were recruited and volunteered as participants (Table 1).” – It seems that the number of participants can be sufficient, but it is mandatory to insert sample size calculation in order to confirm this assumption

Of course, the sample size calculation has been included as suggested.

L97. Please insert if there is maximal age limit to be included in the study; and if this would have affected the results, please provide the corresponding discussion

No, there was no maximal age limit. However, and as you can see from the standard deviation in Table 1, the group was quite homogeneous in terms of age. Not surprisingly, since the participants were recruited among students at the campus.

However, I conducted a study among athletes competing at national and international level with a mean age 34.3, but a standard deviation of 14.1 years. Three of the participants were closer to 60 years while the others were between 25-35 years. Out of curiosity, we examined barbell kinematics. We found no difference between the oldest and the youngest when testing them to failour in the bench press. Whether the same findings would occur in squat and with less trained subjects, is uncertain. However, factors such as lean mass, muscle mass, training status/experience would most likely have a greater impact than age itself (at least to a certain age ?).

L98. The authors should have included at least one reference supporting the technique of dividing participants into below and above average. Please insert/comment in the new manuscript version; I had already heard about the median-split technique before but not average

Wow, thank you. We used the median-split technique but did not know the name before now and just called in above and below average. We now realize that our choice of name was terribly wrong as it would result in a different number of participants in the two groups (20 vs. 26 participants for the above and below average). The median-split technique has been included in the manuscript. Thank you once again.

L158-160. “A visual inspection of the velocity curve was used to identify the different regions and if there was any doubt, a second opinion was gathered.” - Why not implement an automatic detection method (it may already exist)? If the authors choose to keep it this way, it is necessary/mandatory to include measures of intra- and inter-rater variability and ensure that this does not compromise the results of the study.

Thank you for bringing this to our attention. We agree the statement is a bit unclear. To clarify, a visual inspection was conducted to identify the 9th repetition, the end of the descending phase and the end of the ascending phase. Thereafter, we used the raw data (barbell displacement measured with the frequency of 200 Hz per second) to identify the three regions using the first peak acceleration, the lowest acceleration, and the second peak acceleration. We could have used an automatic system to identify these points, but this is quite straightforward when you have the start and end of the ascending phase. The manuscript has been changed accordingly to provide clarity of the procedures. Thank you once again.

L161-162. FIGURE 1 contains orthographical errors, please revise and replace with a figure with better image resolution

We apologize. The figure has been changed accordingly, and we have included the acceleration curve to provide clarity for the general reader.

L186-188. “All participants showed a reduction in acceleration in the sticking region, showing a similar pattern throughout the ascending movement.” – This statement has been based on statistical analysis, if not please consider delete

The sentence has been re-written, and statistics have been included as suggested. Thank you for bringing this to our attention as we have taken this for granted.

L190. “The above average group lifted 0.39 kg (28%, g = 2.61) more per bodyweight compared to the below average group.” – insert the p-value

p- value inserted, thank you

L191. “There was also a strong tendency (p = 0.05, g = 0.60) towards a 6% shorter lifting distance for the above average group (2.07 cm).” – do not declare tendencies

Changed, as suggested

L231. In the first paragraph of discussion, I suggest to repeat the main objective before reporting the main findings

Included, as suggested.

L238. “All participants showed a reduction in acceleration in the sticking region.” – it seems that no results with statistical confirmation have been presented to support such statement in the results section. Please revise the discussion as well as the whole manuscript to avoid making statement that are not based on statistical outputs

Agree, and apologize, the statistics to support our statement have been included throughout the manuscript.

L317. The first sentence of conclusions section is currently awkward, please revise “The present study was that the lifting kinematics . . .” (sic)

The sentence has been rewritten “The main findings of the present study…

Thank you once again for your insight and valuable comments/suggestions.

Round 2

Reviewer 1 Report

Comments and Suggestions for Authors

Writing Revision List

Line 9: Should be written: "Background/Objectives: The barbell back squat is one of the most frequently used exercises to improve lower-body strength and power."
Line 11: Should be written: "Methods: Forty-six recreationally trained men completed five familiarisation sessions over three weeks to ensure proper lifting technique."
Line 13: Should be written: "The participants were tested for ten-repetition maximum (10RM), during which barbell velocity, acceleration, vertical displacement, and the timing of the pre-sticking, sticking, and post-sticking regions were measured."
Line 20: Should be written: "Conclusion: These findings suggest that distinct levels of relative strength may not influence lifting kinematics during 90-degree back squats among recreationally trained participants."
Line 32: Should be written: "However, a shallower squat (e.g., 90-degree knee angle) is recommended for beginners and recreationally trained individuals to reduce injury risk [3,4]."
Line 37: Should be written: "Previous studies have examined factors affecting kinematics, including repetitions to fatigue [1,9,10], loading variations [11,12], elastic bands [13], overload (>1-RM) [7,12], stance width [14], and barbell placement [14]."
Line 41: Should be written: "The most recent study [15] concluded that the sticking region results from reduced force output during the pre-sticking and sticking phases."
Line 44: Should be written: "To the authors’ knowledge, scientific literature examining the influence of relative strength on lifting kinematics in squats is scarce."
Line 50: Should be written: "However, to the authors’ knowledge, the influence of relative strength (i.e., lifted load relative to body weight) on the sticking region has not been examined in squats, though it has been studied in bench press [17,18]."
Line 57: Should be written: "Mauschund and Krosshaug [18] compared bench press technique and joint kinematics between powerlifters (relative 1-RM strength: 1.55) and recreational lifters (relative 1-RM strength: 1.33)."
Line 79: Should be written: "Based on findings from Saeterbakken et al. [17], we hypothesised that the group with greater relative strength would exhibit lower barbell velocity and longer lifting time in the sticking region than the weaker group."
Line 101: Should be written: "All participants were males without pain or injuries that could impair maximal effort during testing."
Line 104: Should be written: "Before experimental testing, squat technique was subjectively evaluated (e.g., stable lumbar-pelvic-hip complex, absence of knee valgus) to ensure safety."
Line 115: Should be written: "Before all familiarisation and testing sessions, participants performed a standardised warm-up with free-weight squats."
Line 132: Should be written: "To measure squat depth, a horizontal band corresponding to each participant’s 90-degree knee angle (measured via goniometer) was positioned behind them."
Line 148: Should be written: "A linear encoder (ET-Enc-02, Ergotest Innovation AS, Porsgrunn, Norway) sampling at 200 Hz with 0.0019 mm resolution was attached to the barbell to analyse kinematics."
Line 170: Should be written: "Statistical analyses were performed using SPSS (Version 29.0, IBM Corp., Armonk, NY)."
Line 197: Should be written: "Velocity (m/s)"
Line 218: Should be written: "Displacement (cm)"
Line 239: Should be written: "When combining data from both groups, a weak negative correlation between relative strength and barbell displacement was observed for the entire movement (Pearson’s r = −0.38, p = 0.01)."
Line 257: Should be written: "The lack of barbell velocity reduction was likely due to the 90-degree knee angle."
Line 284: Should be written: "The above-median group exhibited a non-significant 6% shorter vertical barbell displacement (−2.07 cm, g = 0.60) compared to the below-median group."
Line 287: Should be written: "This difference may be attributable to the 2.5% shorter stature (4.4 cm) in the above-median group (Table 1), potentially implying shorter thigh bones and reduced displacement."
Line 317: Should be written: "Author Contributions: A.H.S. and V.A. conceptualised the study; A.O. performed experiments; V.A. analysed data; A.O. and A.H.S. wrote the manuscript. All authors reviewed and approved the final version."

Scientific and Structural Review List

L. 19: The authors should clarify the phrasing "non-statistical difference (p=0.05)" as p=0.05 indicates a marginal statistical trend. Revise to "non-significant difference (p=0.05)" or justify the terminology.
L. 36: The authors should support the claim "to reduce injury potential" for 90-degree squats with direct citations matching reference [34] (currently [34] is incomplete).
L. 87–88: The authors should justify why the 9th repetition (not the 10th) was analysed, given its proximity to fatigue. Cite prior validation of this approach.
L. 143–144: The authors should correct "Molocity" and "Elioplacement" in Table 2 to "Velocity" and "Displacement" to maintain data credibility.
L. 197–239 (Table 2): The authors should report exact p-values (e.g., p=0.052) for the displacement result (p=0.05) to clarify statistical ambiguity.
L. 257–260: The authors should temper claims about squat depth effects, as studies cited (e.g., [40,41]) compared half/full squats but did not measure sticking regions.
L. 284–288: The authors should acknowledge that limb segment lengths (not just stature) may explain displacement differences and note this as a limitation.
L. 301: The authors should discuss why relative strength did not affect kinematics despite mechanistic theories (e.g., elastic energy transition [42]), contrasting bench press findings [17,18].
L. 317–318: The authors should clarify funding: State "This research received no external funding" in the Funding section if accurate, or remove the placeholder.
L. 325: The authors should omit "patient(s)" in the Informed Consent Statement, as participants were healthy volunteers.

Comments on the Quality of English Language

The manuscript is understandable but requires moderate revisions to meet standards for formal academic publication. While the scientific content is communicated adequately, recurring grammatical inaccuracies, inconsistent terminology, and syntactical issues detract from professionalism.

Author Response

Thank you so much for your help to improve the manuscript including the corrections of sentences/words. All suggestions have been included with the exception of one. The references to SPSS software have not been altered. IBM has gone tired of researchers not citing the IBM SPSS Statistics correctly. On their homepage, they have provided a list of how to cite the current and previous versions of the software. Accordingly, we have used IBM`s preferred version of citing. We have attached the link here

How to cite IBM SPSS Statistics or earlier versions of SPSS

Line 9: Should be written: "Background/Objectives: The barbell back squat is one of the most frequently used exercises to improve lower-body strength and power."
Line 11: Should be written: "Methods: Forty-six recreationally trained men completed five familiarisation sessions over three weeks to ensure proper lifting technique."
Line 13: Should be written: "The participants were tested for ten-repetition maximum (10RM), during which barbell velocity, acceleration, vertical displacement, and the timing of the pre-sticking, sticking, and post-sticking regions were measured."
Line 20: Should be written: "Conclusion: These findings suggest that distinct levels of relative strength may not influence lifting kinematics during 90-degree back squats among recreationally trained participants."
Line 32: Should be written: "However, a shallower squat (e.g., 90-degree knee angle) is recommended for beginners and recreationally trained individuals to reduce injury risk [3,4]."
Line 37: Should be written: "Previous studies have examined factors affecting kinematics, including repetitions to fatigue [1,9,10], loading variations [11,12], elastic bands [13], overload (>1-RM) [7,12], stance width [14], and barbell placement [14]."
Line 41: Should be written: "The most recent study [15] concluded that the sticking region results from reduced force output during the pre-sticking and sticking phases."
Line 44: Should be written: "To the authors’ knowledge, scientific literature examining the influence of relative strength on lifting kinematics in squats is scarce."
Line 50: Should be written: "However, to the authors’ knowledge, the influence of relative strength (i.e., lifted load relative to body weight) on the sticking region has not been examined in squats, though it has been studied in bench press [17,18]."
Line 57: Should be written: "Mauschund and Krosshaug [18] compared bench press technique and joint kinematics between powerlifters (relative 1-RM strength: 1.55) and recreational lifters (relative 1-RM strength: 1.33)."
Line 79: Should be written: "Based on findings from Saeterbakken et al. [17], we hypothesised that the group with greater relative strength would exhibit lower barbell velocity and longer lifting time in the sticking region than the weaker group."
Line 101: Should be written: "All participants were males without pain or injuries that could impair maximal effort during testing."
Line 104: Should be written: "Before experimental testing, squat technique was subjectively evaluated (e.g., stable lumbar-pelvic-hip complex, absence of knee valgus) to ensure safety."
Line 115: Should be written: "Before all familiarisation and testing sessions, participants performed a standardised warm-up with free-weight squats."
Line 132: Should be written: "To measure squat depth, a horizontal band corresponding to each participant’s 90-degree knee angle (measured via goniometer) was positioned behind them."
Line 148: Should be written: "A linear encoder (ET-Enc-02, Ergotest Innovation AS, Porsgrunn, Norway) sampling at 200 Hz with 0.0019 mm resolution was attached to the barbell to analyse kinematics."
Line 170: Should be written: "Statistical analyses were performed using SPSS (Version 29.0, IBM Corp., Armonk, NY)."
Line 197: Should be written: "Velocity (m/s)"
Line 218: Should be written: "Displacement (cm)"
Line 239: Should be written: "When combining data from both groups, a weak negative correlation between relative strength and barbell displacement was observed for the entire movement (Pearson’s r = −0.38, p = 0.01)."
Line 257: Should be written: "The lack of barbell velocity reduction was likely due to the 90-degree knee angle."
Line 284: Should be written: "The above-median group exhibited a non-significant 6% shorter vertical barbell displacement (−2.07 cm, g = 0.60) compared to the below-median group."
Line 287: Should be written: "This difference may be attributable to the 2.5% shorter stature (4.4 cm) in the above-median group (Table 1), potentially implying shorter thigh bones and reduced displacement."

Line 317: Should be written: "Author Contributions: A.H.S. and V.A. conceptualised the study; A.O. performed experiments; V.A. analysed data; A.O. and A.H.S. wrote the manuscript. All authors reviewed and approved the final version."

Thank you once again. Your help is deeply appreciated and, as mentioned earlier, all but one suggestion have been included in the current version of the manuscript. Consequently, we have not commented/replied on each of your comments.

Scientific and Structural Review List

Thank you for taking your time to improve the manuscript by including valuable insight and knowledge. We hope you`ll find the changes and point-to-point answers acceptable

  1. 19: The authors should clarify the phrasing "non-statistical difference (p=0.05)" as p=0.05 indicates a marginal statistical trend. Revise to "non-significant difference (p=0.05)" or justify the terminology.

The sentence has been changed to “There was a 5.86% (2.07 cm) non-statistical difference (p=0.05, g = 0.60) in vertical barbell displacement between the groups” throughout the manuscript.

  1. 36: The authors should support the claim "to reduce injury potential" for 90-degree squats with direct citations matching reference [34] (currently [34] is incomplete).

The sentence has been re-written.

  1. 87–88: The authors should justify why the 9th repetition (not the 10th) was analysed, given its proximity to fatigue. Cite prior validation of this approach.

Thank you for addressing this. Unfortunately, I/we did not find a validation of our approach. The choice was made based on a previous studies (ref 10 and 13). To clarify our point, I have attached the velocity curve of the two final repetitions among one of the subjects (please see the attached file and not the web browser). The two vertical lines are the beginning of the ascending phase. As you can see, there is a negative velocity in the sticking region for the final repetition which is not common. Another example is that the Vmax 2 was lower than Vmax 1 as displaced on the final repletion of the attached figure. Please see the attached figure for further details. In summary, our experience is that the final repetition sometimes/often deviates from the expected/normal velocity curve (which we believe could be explained by a change in effort as the participant know it is their last repetition).   

Furthermore, there are studies demonstrating no differences in barbell kinematics comparing 1,- 3,- 6,- and 10RM and repetition 5 and 6 in 6RM testing (both studies examined squats). We have added these references in the revised manuscript.

  1. 143–144: The authors should correct "Molocity" and "Elioplacement" in Table 2 to "Velocity" and "Displacement" to maintain data credibility.

Changed as suggested

  1. 197–239 (Table 2): The authors should report exact p-values (e.g., p=0.052) for the displacement result (p=0.05) to clarify statistical ambiguity.

We understand the possible statistical ambiguity. All statistical differences are marked with the symbol * in table 2 and we have included 0.052 to avoid confusion. With the re-writing of the sentence “non-statistical difference (p=0.05)” throughout the manuscript, we hope we have clarified potential ambiguity.

  1. 257–260: The authors should temper claims about squat depth effects, as studies cited (e.g., [40,41]) compared half/full squats but did not measure sticking regions.

Your point has been included in the discussion.

  1. 284–288: The authors should acknowledge that limb segment lengths (not just stature) may explain displacement differences and note this as a limitation.

We acknowledge your comment and your point has been included in the discussion.

  1. 301: The authors should discuss why relative strength did not affect kinematics despite mechanistic theories (e.g., elastic energy transition [42]), contrasting bench press findings [17,18].

We have expanded the discussion, but moved it to the previous paragraph where the elastic potential and F-V profile between strength levels are disused.

  1. 317–318: The authors should clarify funding: State "This research received no external funding" in the Funding section if accurate, or remove the placeholder.

Changed, as suggested. This research received no external funding.

  1. 325: The authors should omit "patient(s)" in the Informed Consent Statement, as participants were healthy volunteers.

Changed to “participant(s).

Reviewer 2 Report

Comments and Suggestions for Authors

Thank you for taking into account my previous suggestions in the revised manuscript version. However, some points still requires attention. My major concern refers to visual analysis as part of the methods. The authors should to insert the requested intra- and inter-rater agreement measures in order to explicit that the method can be suitable for scientific purposes and does not compromise the findings. Please see some specific comments that remains open:

P1L19. “There was a 6% shorter non-statistical difference . . .” - if there was no significant difference, then it is not possible to say that it was shorter. – ALREADY WRITTEN IN THE FIRST REVIEW

P3L82. To the authors' best knowledge, no previous studies have examined how strength levels influence the kinematics during the back squat in beginners.  - It is important to state if this justification has been based on a systematic review or something similar so that the justification for the study can be much more solid. – ALREADY WRITTEN IN THE FIRST REVIEW

P5L192-198. It is necessary/mandatory to include measures of intra- and inter-rater variability and ensure that this does not compromise the results of the study. – ALREADY WRITTEN IN THE FIRST REVIEW

Author Response

Thank you for taking into account my previous suggestions in the revised manuscript version. However, some points still requires attention. My major concern refers to visual analysis as part of the methods. The authors should to insert the requested intra- and inter-rater agreement measures in order to explicit that the method can be suitable for scientific purposes and does not compromise the findings. Please see some specific comments that remains open:

Thank you for taking your time to improve the manuscript by including valuable insight and knowledge. We hope you`ll find the changes and point-to-point answers acceptable

P1L19. “There was a 6% shorter non-statistical difference . . .” - if there was no significant difference, then it is not possible to say that it was shorter. – ALREADY WRITTEN IN THE FIRST REVIEW

The sentence has been changed to “There was a 5.86% (2.07 cm) non-statistical difference (p=0.05, g = 0.60) in vertical barbell displacement between the groups” throughout the manuscript.

P3L82. To the authors' best knowledge, no previous studies have examined how strength levels influence the kinematics during the back squat in beginners.  - It is important to state if this justification has been based on a systematic review or something similar so that the justification for the study can be much more solid. – ALREADY WRITTEN IN THE FIRST REVIEW

A literature search was performed using PubMed, SPORT Discus, Web of Science, and Google Scholar databases. The topic was systematically searched using a Boolean search strategy with the operators AND, OR, NOT and a combination of the following keywords: (“resistance training” OR “weight training” OR “power training” OR “squat*” OR “bench press” OR “power lifting”) AND ( “relative strength” OR “strength levels” OR “advanced” OR “beginners” OR “expert” OR “recreational”)  AND “biomechanics” OR “kinematic” OR “sticking region” OR “sticking point” OR “barbell velocity” OR “barbell acceleration”) NOT (“patient” OR “rehabilitation” OR “injuries” OR “clinical”). The identified papers were merged into one database to eliminate duplications. All references were manually crosschecked to identify potential studies that might have been missed in the first search. Due to the short time frame (10 days), only one author conducted the title and abstract screening. The relevant papers (n = 2) have been included in the manuscript (in addition to two papers already included) to strengthen the novelty of the present study.

P5L192-198. It is necessary/mandatory to include measures of intra- and inter-rater variability and ensure that this does not compromise the results of the study. – ALREADY WRITTEN IN THE FIRST REVIEW

The intra- and inter- rater variability (ICC) has been included as suggested and discussed as a limitation in the discussion.

Round 3

Reviewer 2 Report

Comments and Suggestions for Authors

The authors have responded to all my questions. Thank you.